# A Population-Based Analysis of the Cancer Incidence in Individuals under 50 in a Northern Italian Province: Focusing on Regional Disparities and Public Health Implications

**DOI:** 10.3390/ijerph21101333

**Published:** 2024-10-08

**Authors:** Lucia Mangone, Francesco Marinelli, Isabella Bisceglia, Francesca Roncaglia, Fortunato Morabito, Cinzia Testa, Carmine Pinto, Antonino Neri

**Affiliations:** 1Epidemiology Unit, Azienda USL-IRCCS di Reggio Emilia, 42122 Reggio Emilia, Italy; francesco.marinelli@ausl.re.it (F.M.); isabella.bisceglia@ausl.re.it (I.B.); francesca.roncaglia@ausl.re.it (F.R.); 2Gruppo Amici Dell’Ematologia Foundation-GrADE, 42122 Reggio Emilia, Italy; f.morabito53@gmail.com; 3Sala Stampa Nazionale—Milano, 20100 Milano, Italy; cinziatesta.studio@gmail.com; 4Medical Oncology Unit, Azienda USL-IRCCS di Reggio Emilia, 42122 Reggio Emilia, Italy; carmine.pinto@ausl.re.it; 5Scientific Directorate, Azienda USL-IRCCS di Reggio Emilia, 42122 Reggio Emilia, Italy; antonino.neri@ausl.re.it

**Keywords:** cancer trends, regional disparities, public health strategies, epidemiology, screening practices, risk factors, young cancer

## Abstract

International studies have shown an increase in cancer incidence among young adults, raising public concern. This study aims examines trends in the cancer incidence among individuals aged 15–49 years in a province of Northern Italy, covering diagnoses from 1996 to 2021, and compares the annual percentage change (APC) with national and international data. In males, the overall cancer incidence showed a modest increase between 1996 and 2013 (APC 1.6), followed by a decline in the subsequent years (APC −2.5). In females, there was a modest increase over the entire period (APC 1.0). The lung cancer incidence decreased in both sexes (APC −3.9 in males and APC −3.3 in females), while a decrease was observed for colorectal cancers in women (APC −2.4). Since 2015, the thyroid cancer incidence declined significantly in females (APC −10.2), while an increase was noted in males (APC 2.5). The testicular cancer incidence rose in males (APC 1.5), and the melanoma incidence increased in both sexes (APC 3.4 in males and APC 3.9 in females). The breast cancer incidence remained stable (APC 0.3). These results underline the importance of promoting healthy lifestyles even among younger generations to address emerging cancer trends and support cancer prevention efforts.

## 1. Introduction

Previous studies have suggested that the incidence of cancer at different tumor sites in adults under the age of 50 has been rising in many parts of the world since the 1990s [1,2,3]. Globally, cancer accounted for approximately 20 million new cases in 2022 [4], with an age-standardized rate (ASR) of 196.9 × 100,000 for all ages; among these, 3.4 million cases are reported in individuals under the age of 50 (ASR 51.3 × 100,000) [5].

Although the cancer incidence predominantly rises in individuals over 50, a 2022 review highlights that an increasing trend is observed in those under 50 [3], particularly for cancers of the breast, colorectal, endometrium, bladder, and esophagus. This increase could largely be attributed to early diagnostic practices and the availability of screening programs, as well as a growing exposure to risk factors during youth [6].

Enhanced participation in cervical cancer screening among women aged 21 to 65 [7] and the advancement of colorectal cancer screening to begin at age 45, as already implemented in the USA [8], may have contributed to the observed rise in cancer cases among the younger population. For breast cancer, in many countries, the recommendation to perform a mammogram every 1–2 years between the ages of 40–49 has led to an increase in breast cancer diagnoses [9,10,11]. In addition to early diagnosis, lifestyle changes such as unhealthy diets, sedentary behavior leading to increased obesity, and environmental pollution are significant contributors to the rising cancer incidence in younger individuals [12]. Other notable risk factors include high alcohol consumption, smoking, and a decline in pregnancy rates among young women [2,3]. A recent analysis of 29 different tumor sites in young people highlighted a growing increase in tumor pathology, with substantial variability between sex, age groups, countries, and tumor sites [13].

More recently, a global study focusing exclusively on subjects under 50 [1] revealed a similar upward trend in early-onset cancers, with detailed analyses by tumor sites and geographic disparities. In particular, this publication indicated a 79.1% increase globally in youth cancers between 1990 and 2019, with projections to 2030 indicating an increase in incidence of 31% and mortality of 21%.

These numbers have deeply alarmed public opinion, triggering not only scientific debate but also a growing interest that has involved the general population and the media, raising a worrying debate in the public opinion of our country [14,15]. In light of this broad debate, we have decided to focus our attention on the incidence of tumors and the trend in subjects under 50 years of age in our population, trying to understand if these results are confirmed and identifying the possible causes.

The objective of this work is to study the incidence of tumors in patients under 50 in a province of Northern Italy, covered by a cancer registry, using incident data in the period of 1996–2021.

## 2. Materials and Methods

This study focused on cancer cases occurring in the province of Reggio Emilia, Northern Italy, which has a population of more than 590,000 inhabitants. This area is noteworthy, not only for the presence of a well-established cancer registry but also for the presence of an important Comprehensive Clinical Cancer Center (IRCCS of Reggio Emilia) accredited by the OECI.

The province of Reggio Emilia is characterized in general by a high incidence of tumors (like all the provinces of Northern Italy) but also by a high quality of care, which guarantees high survival rates and high compliance with oncological screening [16,17].

### 2.1. Data Source

The data were obtained from the Reggio Emilia Cancer Registry, which uses the classification by topography and morphology of the ICD-O III edition [18]. The study population consisted of patients aged 15 to 49 years (221,846 cases) in the period of 1996–2021. This study evaluated the overall incidence of infiltrating malignant tumors and the more frequent tumor sites in young people. Descriptive analyses and incidence trends were evaluated by sex, age, and tumor site.

The incidence trends, calculated in the 1996–2021 period, were reported for both subjects under 50 and over 50. For two major cancer screening targets (colorectal and breast cancer), trends were reported for the ages< 45, 45–49, 50–69, and 70.

A summary table of the trends in the province of Reggio Emilia shows trends by site and gender. The 9 most frequent tumor sites showing a significant increase at an international level (nasopharynx, prostate, thyroid, kidney, colorectal, skin, uterus, testis, and pancreas) [1] were compared with national and local data; the 3 most frequent tumor sites in Italy (melanoma, lung, and breast) were added to these sites [19].

### 2.2. Data Analysis

Standardized incidence rates for the period of 1996–2021 were calculated for all cases, and specific tumor sites were divided by gender. For age groups, specific incidence rates were calculated using the population of the province of Reggio Emilia provided by the Italian National Institute of Statistics (recorded on 1 January each year) as denominators. The direct method was applied to standardize the incidence rates (calculated per 100,000 person-years), using the 2013 European Standard Population as a reference [20]. Trends over time were analyzed by calculating the annual percentage change (APC) in age-standardized rates using joinpoint regression. Joinpoint regression is a methodology used for estimating trends over time in cancer studies. The SEER Cancer Statistics Review (CSR) uses the same Joinpoint Regression Program [21]. APC is one way to characterize trends in cancer rates over time by estimating the annual percentage rate change.

We used a segmented log-linear regression. With this approach, the cancer rates are assumed to change at a constant percentage of the rate of the previous year. One advantage of characterizing trends this way is that it is a measure that is comparable across scales for both rare and common cancers. It is not always reasonable to expect that a single APC can accurately characterize the trend over an entire series of data. The joinpoint model uses statistical criteria to determine when and how often the APC changes. For cancer rates, it is fit using joined log-linear segments, so each segment can be characterized using an APC. Finding the joinpoint model that best fits the data allows us to determine how long the APC remained constant and when it changed. The maximum number of joinpoints allowed for these analyses was four [22]. Statistically significant APC values in Reggio Emilia were reported and compared with respective APCs recorded in Italy [19] and globally [1]. Analyses were performed using Stata 16.1 SE (Stata Corp, College Station, TX, USA).

## 3. Results

In the period of 1996–2021, the province of Reggio Emilia recorded 11,102 cancer cases in patients aged 15–49 years. The distribution of cases by age, sex, and tumor site is detailed in Table 1. The average age at diagnosis was 40,6 years, with a higher prevalence in females, accounting for 62.5% of the cases. Breast cancer emerged as the most common type, representing 21% of all neoplasms, followed by thyroid, melanoma, colorectal, testicular, and lung cancers. Table 2 shows the incidence of the five most frequent tumor sites, categorized by sex and age groups. In young males (15–24 years) Hodgkin’s lymphoma is the most frequent tumor, followed by the testis; between the ages of 25–34, the testis is the most frequent tumor, followed by the thyroid; after the age of 35 years, melanoma is the most frequent tumor, followed by the thyroid in the age group of 35–44 and Non-Hodgkin Lymphoma in the age group of 45–49. Among young females, thyroid tumors are the most frequent neoplasms up to the age of 34. However, from the ages of 35–44 years and 45–49 years, breast cancer becomes the most frequent tumor, with prevalence rates of 34% and 43%, respectively.

By analyzing the incidence trends over the period of 1996–2021, we observed distinct patterns across different cancer types (Figure 1). For all tumors combined, the trend in males (Figure 1A) showed a significant increase from 1996 to 2013 with an APC of 1.6, followed by a marked decline (APC −2.5), while in women (Figure 1B), there was a slight increase over the entire period (APC 1.0). Breast cancer (Figure 1C) exhibited a slight increase across the study period (APC 0.3), while testicular cancer (Figure 1D) demonstrated a noteworthy rise (APC 1.5). Lung cancer (Figure 1E) showed a significant decline both in males (APC −3.9) and females (APC −3.3). The colorectal cancer trends (Figure 1F) varied by gender: in males, the incidence remained stable (APC −0.2), whereas females showed a substantial decline (APC −2.4). Thyroid cancer (Figure 1G) increased in males (APC 2.5), while in females, the incidence increased significantly until 2014 (APC 5.6), followed by a significant decrease in the following period (APC −10.2). The melanoma incidence (Figure 1H) displayed a substantial increase in both genders, with a similar value in males (APC 3.4) and females (APC 3.9). A summary of the trends in the listed tumor sites is shown in Table 3.

Figure 2 illustrates trends in the breast and colorectal cancer incidences by age group. For breast cancer (Figure 2A), the trend remains stable up to the age of 44; in the 45–49 age group, a slight increase was observed in 2010, probably related to the expansion of the screening target population. In the age groups of 50–69 years (screening interval) and 70+, a decreasing trend was observed, with only a slight increase in 2021, following the resumption of screening activities after the suspension during the COVID-19 emergency. For colorectal cancer (Figure 2B), the trend is stable in the younger age group but shows a sharp decline in the 50–69 age group (screening interval) and in those who were 70 and above.

The incidence rates for the respective tumor sites in individuals over 50 years of age are detailed in Appendix A. In patients over 50, the tumor incidences decrease overall in both males (APC −1.8) and females (APC −1.0); furthermore, thyroid tumors decrease in males (APC −2.0 *), but lung tumors increase in females (APC +2.8).

Finally, Table 4 summarizes a comparison of the data observed in the province of Reggio Emilia with national and international data. Compared to the tumor sites for which a significant increase has been reported worldwide, in Italy, an increase in prostate, thyroid, kidney, uterus, and testicular tumors is confirmed. In these tumor sites, in Italy, there is also an increase in melanomas and breast tumors and a decrease in lung tumors in males. The national results are also confirmed by our study, which also adds a decline in colorectal cancers.

## 4. Discussion

This study aimed to analyze the temporal trends in the cancer incidence among patients under the age of 50 in the province of Reggio Emilia, Northern Italy, and to compare the results with national and international outcomes. The data encompass over 11,000 cases, providing a robust foundation for understanding local trends in cancer incidence.

The incidence trend for all tumor sites in our study shows an increase in males until 2013 (APC 1.6) followed by a decline in subsequent years, while in females, there is an increase of 1% per year; the data are comparable to what was observed in Italy in both males (0.7%) and females (1.6%). Zhao’s study [1] identifies nine tumor sites with a statistically significant increase in incidence. We compared these international findings with data from Italy and our study results. The first tumor site, the nasopharynx, shows a significant increase (APC 2.3), which is not observed either in Italy or in our data. This difference could be explained by a lower presence of known risk factors such as tobacco smoking and air pollution but also HPV infection, which is responsible for 75% of these tumors [23,24].

The incidence of prostate cancer is increasing worldwide (APC 2.2) [1], including in Italy (APC 3.4) [19], but it remains low in Reggio Emilia, where only 37 cases were recorded in the entire study period. The increase in incidence globally has been confirmed by other international studies, underlining the impact of rigorous guidelines for PSA screening [25,26,27,28,29]. In Reggio Emilia, the indications for the use of PSA are very stringent, with an initial decline in cases even among the older age groups. *Thyroid cancer* also presents an interesting trend, characterized by a global APC of 1.9, which was confirmed in Italy by an increase in females (APC 3.5), in stark contrast to the significant decline observed in Reggio Emilia starting from 2014 (−10.2). The incidence of this neoplasm is significantly influenced by advances in diagnostic methods [30], which have led to an overall increase in cases detected in various age groups worldwide [31]. In our province, public health policies aimed at curbing unnecessary diagnostic interventions have been effective in controlling inappropriate diagnoses.

Kidney cancer, although showing a significant global increase (APC 1.7), which in Italy was confirmed only in males (APC 2.5), remained relatively rare in Reggio Emilia: also, in this case, the widespread use of advanced imaging modalities has likely contributed to an increase in the incidental detection of kidney cancer [32,33], which is unconfirmed in our province.

Colorectal cancer shows an increasing global trend among young people (APC 1.7), who often present at more advanced stages than older adults [34]. In Italy, the trend remains stable according to AIOM data [19], while in Reggio Emilia, a decline in incidence is observed, which appears to be statistically significant in women. While the risk factors that are responsible for the increase are well known [35,36], the decline in incidence is largely attributable to the positive impact of screening programs [37,38]. Although colorectal cancer is primarily targeted at subjects aged 50–69, beneficial effects have also been reported among younger populations in the study areas [39,40]. This suggests that awareness and early detection strategies may have broader implications beyond the targeted age group [41].

Non-melanoma skin cancers have shown a global increase of 1.6%, which is consistent with other international studies [42]. National and regional data are not available, as non-melanoma skin cancers are typically excluded from the international calculations and comparisons. However, in Reggio Emilia, their incidence has risen from 32 in 1996 to 81 cases in 2021. This increase is largely attributed to higher exposure to risk factors such as ultraviolet radiation and improved early diagnostic practices [43].

The uterine cancer trends, showing a global increase (APC 1.3) [12] and a marked rise in Italy (APC 2.3) [18], are also significant. Notably, young women have experienced the most pronounced increase, attributed to factors such as obesity and family history [44,45,46,47]. These increases are not reported in our province, where uterine cancer remains rare under the age of 50.

The rising incidence of testicular cancer globally (APC 1.2) [12], reflected in Italy (APC 2.6) [18] and Reggio Emilia (APC 1.5), aligns with known risk factors like cryptorchidism and family history [48,49]. This increasing trend connects to broader patterns observed in various cancer types and suggests that further investigation into risk factors and their regional variations is necessary [50]. Pancreatic cancer, which shows a global increase (APC 1.1) [18] that was confirmed in another study [51], is not reflected in Italian or regional data. This disparity suggests variations in diagnostic practices or healthcare access that warrant further investigation [52].

Three other tumor sites that do not show significant increases worldwide were taken into consideration, because they show significant increases in Italy.

Melanoma, which globally shows only a slight increase (APC 0.7), instead shows a marked increase in Italy (APC 7.3 in males and 7.6 in females), which is also confirmed by the data from Reggio Emilia. The increase in melanoma in young people [53] seems to be linked to sunburn during childhood and adolescence [54,55]. However, the increase in incidence (the SEER data also confirm an increase in both young males and females) is associated with a decrease in mortality linked to both early diagnosis and adequate treatment [56].

Lung cancer shows a significant decline worldwide (APC −0.7) [1], particularly in males [57]. A study conducted in 40 countries worldwide [58] showed a constant decline in men and greater variability in women. The decline in lung cancer in young males, also confirmed in Italy (APC −3.8) [19], seems largely to be linked to the decline in cigarette consumption among young people [59]. More interesting is the fact that there was also a significant decrease in lung cancer in Reggio Emilia among women (APC −3.3). In this case, awareness campaigns aimed at reducing cigarette consumption were implemented by the local health authority (in collaboration with Lega Tumori and Luoghi di Prevenzione) and have probably had a positive outcome.

Breast cancer (BC) shows a non-significant increase globally (APC 1.0) and a significant increase in Italy (APC 1.6), while in our experience, the trend appears stable (APC 0.3). Globally, the increase in breast cancer affects both the age groups of 20–29 (APC 3.1) and 30–39 years (APC 1.2), and in recent years, also ages 40–49 (APC 0.7) [60]. Because there are no screening programs under the age of 40, changes in BC incidence in these women would not be related to screening practices. Rather, the increasing BC rates may reflect an interaction between lifestyle, reproductive habits, environmental factors, and genetic predispositions [61,62]. Early menarche, the use of oral contraceptives, nulliparity, a late age at first birth, and decreased breastfeeding may have contributed to increasing the incidence of BC in both pre- and post-menopausal women [63]. The increase in incidence among women over 40 could be influenced by screening activities [64]. In the USA, the expansion of the target population has led to an increase in incidence even among younger groups [65].

In summary, this work is inspired by the strong emotional impact created in Italy by the publication of Zhao’s article, continuously reported by the national press. However, the focus on tumors in young people and children is not new in Italy, where in the past, other studies have documented attention on these issues. Recent work conducted in Sicily on the cancer trends in children and young adults did not show a significant increase compared to national data [66]. The availability of the specialized cancer registry also seems capable of increasing the level of sensitivity in data collection [67].

Significant attention is also paid in our country to aspects linked to environmental pollution, so much so that there is a study financed by the Ministry of Health (Studio Sentieri) aimed at evaluating the excess risk of cancer in children and adolescents residing near contaminated sites [68]. Similar results in children were also observed for residents in the province of Taranto, characterized by the presence of steel mills, which caused an excess of tumors in children [69].

Concerning our study, we are well aware of the limited area under study, as well as the fact that the lack of information on the stage does not allow for further hypotheses on possible diagnostic advances and that there would be a need for comparison with more recent national data. However, we would like to point out that this study is based on the use of real data relating to 25 years of incidence in an important cancer registry in Northern Italy.

## 5. Conclusions

In conclusion, we have not observed an increase in cancers among young people to justify the alarmism that has arisen at a national level in recent months, following the disclosure of international studies. However, this result must not lead to a decline in primary and secondary prevention campaigns, which are also fundamental in young subjects.

Public health interventions are needed to reduce the habit of smoking, provide information on correct nutrition, and encourage physical activity without neglecting the importance of social relationships. Reggio Emilia’s insights provide a valuable case study for other regions facing similar public health challenges. By leveraging these findings and implementing targeted prevention and intervention strategies, giant strides can be made in reducing the global burden of cancer.

## Figures and Tables

**Figure 1 ijerph-21-01333-f001:**
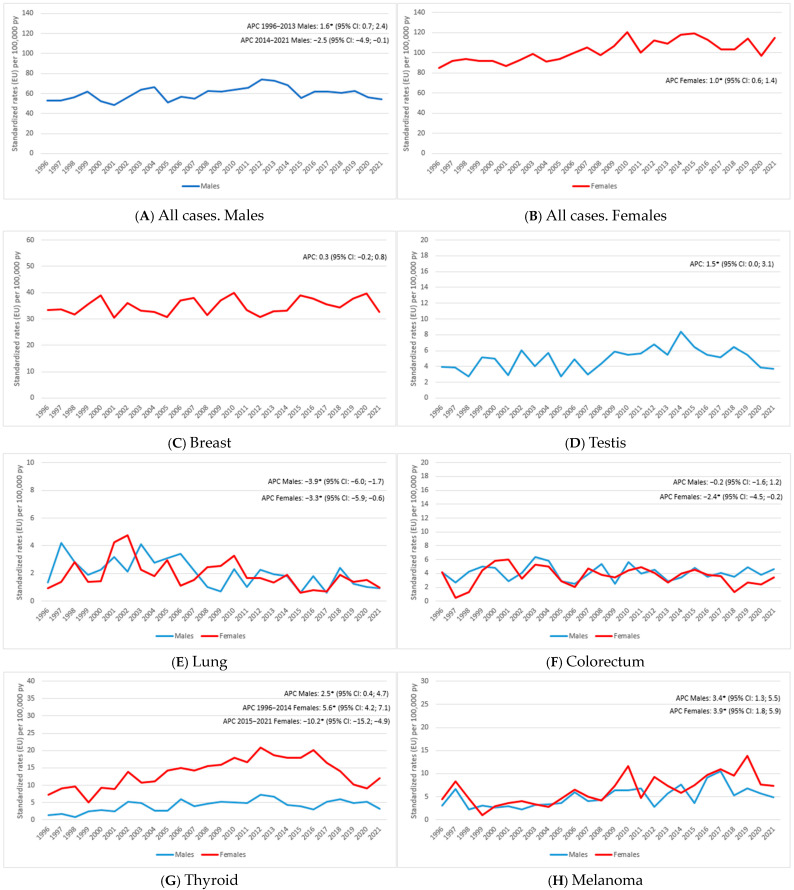
Reggio Emilia Cancer Registry. Years of 1996–2021. Incidence trends in the under-50 population for all cancers for males (**A**) and females (**B**) and in the breast (**C**), testis (**D**), lung (**E**), colorectum (**F**), thyroid (**G**), and melanoma (**H**). * statistically significant.

**Figure 2 ijerph-21-01333-f002:**
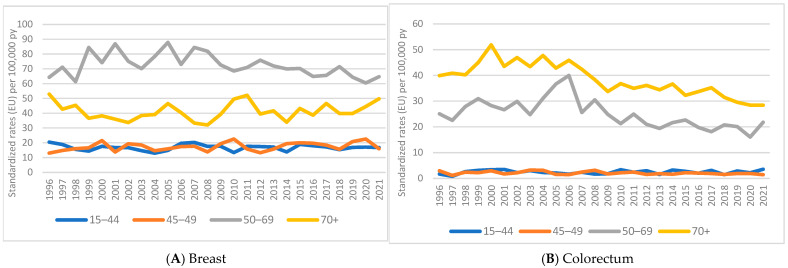
Reggio Emilia Cancer Registry. Years of 1996–2021. Incidence trends in breast (**A**) and colorectal cancer (**B**), divided by age groups.

**Table 1 ijerph-21-01333-t001:** Reggio Emilia Cancer Registry. Years of 1996–2021. Numbers of cancer patients aged 15–49 years, divided by gender and tumor site.

	Reggio	
Population (15–49 years)	221,846	
	mean	sd
Age at diagnosis	40.6	7.4
	n	%
Gender		
Male	4168	37.5
Female	6934	62.5
Site		
Breast	2378	21.4
Thyroid	1226	11.0
Melanoma	805	7.3
Colon–Rectum	531	4.8
Testicular	334	3.0
Lung	266	2.4
Other sites	5562	50.1
Total	11,102	100

**Table 2 ijerph-21-01333-t002:** Reggio Emilia Cancer Registry. Years of 1996–2021. Distribution of the five most frequent cancer incidences as a percentage of the total cancer incidence, stratified by gender and age groups.

	Males				Females			
	15–24	25–34	35–44	45–49	15–24	25–34	35–44	45–49
1	HL 18%	Testis 20%	Melanoma 10%	Melanoma 10%	Thyroid 24%	Thyroid 26%	Breast 34%	Breast 43%
2	Testis 16%	Thyroid 10%	Thyroid 8%	NHL 7%	HL 12%	Breast 16%	Thyroid 14%	Thyroid 8%
3	NHL 8%	Melanoma 10%	Testis 7%	Colon 5%	Melanoma 8%	Melanoma 11%	Melanoma 7%	Melanoma 4%
4	Thyroid 7%	HL 8%	NHL 7%	Lung 5%	NHL 8%	HL 5%	Cervix 3%	Colon 3%
5	Melanoma 6%	NHL 7%	Kidney 4%	Kidney 5%	Brain 6%	Cervix 4%	NHL 3%	Uterus 3%

**Table 3 ijerph-21-01333-t003:** Reggio Emilia Cancer Registry. Incidence cancer trends in 1996–2021. ↑—Statistically significant increase; ↓—statistically significant decrease; ↔—stable trend.

	Males	Female
All	↔	↑
Breast		↔
Lung	↓	↓
Colon	↔	↓
Thyroid	↑	↑
Thyroid (2015–2021)		↓
Testis	↑	
Melanoma	↑	↑

**Table 4 ijerph-21-01333-t004:** Reggio Emilia Cancer Registry. Years of 1996–2021. Numbers of cancer patients aged 15–49 years, focusing on the main tumor sites. Comparison of data with international (1) and national data (2).

	World (1)	Italy (2)		Reggio Emilia	
	M + F	Males	Females	Males	Females
Nasopharynx	**2.3**	0		na	na
Prostate	**2.2**	**3.4**		na	
Thyroid	**1.9**	0	**3.5**	**2.5**	**5.6** (1996–2014) **−10.2** (2015–2021)
Kidney	**1.7**	**2.5**	0	na	na
Colorectal	**1.7**	0	0	−0.2	**−2.4**
Skin	**1.6**	0	0	na	na
Uterine	**1.3**		**2.3**		na
Testis	**1.2**	**2.6**		**1.5**	
Pancreas	**1.1**	0	0	na	
Melanoma		**7.3**	**7.6**	**3.4**	**3.9**
Lung		**−3.8**	0	**−3.9**	**−3.3**
Breast			**1.6**		0.3

Statistically significant data are in bold.

## Data Availability

The data presented in this study are available on request from the corresponding author. The data are not publicly available due to ethical and privacy issues; requests for data must be approved by the Ethics Committee after the presentation of a study protocol.

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
