# Peer review of "A Population-Based Analysis of the Cancer Incidence in Individuals under 50 in a Northern Italian Province: Focusing on Regional Disparities and Public Health Implications"

_ijerph, 2024, doi:10.3390/ijerph21101333_

Round 1

Reviewer 1 Report

Comments and Suggestions for Authors

The main question addressed by the research group is to investigate the cancer youth statistics in Northern part of Italy. Many other researchers are investigating on cancer in Southern Italy, but this group focuses on Northern Italy which makes it relevant for the field. The abstract could be rewritten. In the introduction, why specifically Italy was studied for cancer in youths could be highlighted better. Also, the introduction lacks the objectives or goal of this research study and how it will be done. The figures and graphs need some formatting. The methodology needs more description of the statistics used. The conclusion doesn’t cover the entire results, and it needs to be more descriptive. What are the specific cancer types with increasing or decreasing incidence rates in Northern Italy? One of the main question ‘How do the incidence rates compare to national and international trends’ needs to be highlighted in the conclusion. The references seem to be fine. Are there emerging research questions or new methodologies that could provide additional insights?

Author Response

Comments and Suggestions for Authors

The main question addressed by the research group is to investigate the cancer youth statistics in Northern part of Italy.

Many other researchers are investigating on cancer in Southern Italy , but this group focuses on Northern Italy which makes it relevant for the field.

RE: Thanks to the reviewer. We have added some notes regarding work done in the regions of central and southern Italy on children and young people.

1.The abstract could be rewritten.

RE: Thanks to the reviewer for the suggestion. Since we revised much of the work, we also rewrote the abstract.

  1. In the introduction, why specifically Italy was studied for cancer in youths could be highlighted better. Also, the introduction lacks the objectives or goal of this research study and how it will be done.

RE: Thanks for the suggestion. We have revised the introduction and better explained the objectives of our work.

3.The figures and graphs need some formatting.

RE: We have arranged the figures and tables to make them easier to read.

  1. The methodology needs more description of the statistics used.

RE: Thanks for your comment. We added more information in the data analysis section.

  1. The conclusion doesn’t cover the entire results, and it needs to be more descriptive. What are the specific cancer types with increasing or decreasing incidence rates in Northern Italy?

One of the main question ‘How do the incidence rates compare to national and international trends’ needs to be highlighted in the conclusion. The references seem to be fine.

RE: We have added a table to facilitate a quick view of the tumor sites that are increasing and decreasing, compared with the Italian data. We have better emphasized in the conclusions the tumor sites that require greater attention.

  1. Are there emerging research questions or new methodologies that could provide additional insights?

RE: Thanks to the reviewer for the suggestion. We tried to highlight what could be emerging issues that could interest our territory among young adults.

Reviewer 2 Report

Comments and Suggestions for Authors

The authors have a comprehensive long-term data on a variety of cancer incidences, but unfortunately I feel that the analysis fails to do it justice. The choice of methods is not clearly explained, the results are simply listed rather than summarised, and the discussion is mostly speculation, which makes no effort to look at the trends as a whole. I feel that much more work is needed to make sense of the data.

The authors have used a joinpoint regression, which I find a rather surprising choice. On the one hand, it is certainly not a common modeling choice, and thus needs a bit more justification. On the other hand, there is very little discussion of the modeling results. A joinpoint regression is supposed to be able to identify the change points, i.e., time points when the trend changes. As such, it is interesting to consider why a trend would change (for example, introduction of new screening measures, public health campaigns, social media trends, new medicines etc.) I feel that this should be expounded on.

What do asterisks next to estimates mean? 

I really could not see how the plotted incidences correspond to the estimated APCs. Perhaps, plotting on a logarithmic scale, and adding fitted lines is a good idea? Also, please add the y-axis label and units.

The results section seems just to be a list of things which should be in a table. (That will also make it easier to see at a glance what increased and what decreased). It is tedious to read, and there doesn't seem to be much synthesis there. Placing all the results in the table would make it easier to see the increases and decreases at a glance.

The discussion seems to be as disjointed as the rest of the manuscript. It is often unclear how the references support the argument. For example, the authors say "The increase among young individuals who frequently tan highlights the need for preventive measures [51]." The referenced paper does not seem to contain anything about tanning. They do mention lower odds of survival for 15-25 year olds, but from the analysis here, it is not clear whether the increase observed in this age group occurred in that age group or older. It is also not clear whether the authors ascribe the increase in tanning or something else. Regarding the observed decline in thyroid cancer rates, the authors comment: "This decline may be linked to local efforts to reduce unnecessary diagnostic procedures, highlighting the effectiveness of regional public health measures in controlling cancer rates." But, if so, why isn't this observed in the rates for other cancers?

The abstracts needs to be rewritten. The authors say 

"Our results indicate that young cancers affect 63% of women."

And then the immediate next sentence is 

"In particular, we observed in males a modest increase in the period 1996-2013 (APC 1.6*)"  

This reads like a complete non sequitur. The abstract goes on to list various trends, and then there is an abrupt conclusion that "These results underscore the importance of promoting healthy lifestyle behaviors among younger generations.", which, again, does not seem to follow from anything said before. 

Also, in Line 1 of the Introduction, the authors say: "...cancer accounts for approximately 20 million new cases ..." Should this be "annually"? And again, 197x100,000 per year?

Comments on the Quality of English Language

see comments to the authors

Author Response

Comments and Suggestions for Authors

The authors have a comprehensive long-term data on a variety of cancer incidences, but unfortunately I feel that the analysis fails to do it justice.

1.The choice of methods is not clearly explained

RE:Thanks for your comment. We added more information in the data analysis section.

The results are simply listed rather than summarised, and the discussion is mostly speculation, which makes no effort to look at the trends as a whole. I feel that much more work is needed to make sense of the data.

RE: We have fixed the part of the results that have been summarized and we have also added to the discussion some hypotheses that could support the reasons for the trends in our province.

  1. The authors have used a joinpoint regression, which I find a rather surprising choice. On the one hand, it is certainly not a common modeling choice, and thus needs a bit more justification. On the other hand, there is very little discussion of the modeling results. A joinpoint regression is supposed to be able to identify the change points, i.e., time points when the trend changes.

RE: Thanks for your comment. We used joinpoint regression because is a methodology used for estimates trend over time in cancer study. The SEER Cancer Statistics Review (CSR) uses the same Joinpoint Regression Program.The Joinpoint statistical software was developed by IMS, Inc. under contract to the National Cancer Institute.

  1. As such, it is interesting to consider why a trend would change (for example, introduction of new screening measures, public health campaigns, social media trends, new medicines etc.) I feel that this should be expounded on.

RE: Thanks to the reviewer for the request. We tried to better explain, site by site, why we observed these changes and what the causal events might have been.

  1.  What do asterisks next to estimates mean?  

RE: The asterisk means the statistical significance (p<0.05) but they have been removed.

  1. I really could not see how the plotted incidences correspond to the estimated APCs. Perhaps, plotting on a logarithmic scale, and adding fitted lines is a good idea? Also, please add the y-axis label and units.

RE: Thanks for your comment. APC is one way to characterize trends in cancer rates over time. The logarithmic transformation used in the input file tab to estimate the APC is ln(y)=xb. With this approach, the cancer rates are assumed to change at a constant percentage of the rate of the previous year. One advantage of characterizing trends this way is that it is a measure that is comparable across scales, for both rare and common cancers. For example, it is reasonable to think that rates for a rare cancer and a common cancer could both change at 1% per year, but it is not reasonable to think that a rare cancer and a common cancer would change in the same increments on an absolute (or arithmetic) scale. That is, a cancer with a rate of 100 per 100,000 could be changing by 2 per 100,000 every year, but a cancer with a rate of 1 per 100,000 would probably not change in the same increments.It is not always reasonable to expect that a single APC can accurately characterize the trend over an entire series of data. The joinpoint model uses statistical criteria to determine when and how often the APC changes. For cancer rates, it is fit using joined log-linear segments, so each segment can be characterized using an APC. For example, cancer rates may rise gradually for a period of several years, rise sharply for several years after that, and then drop gradually for the next several years. Finding the joinpoint model that best fits the data allows us to determine how long the APC remained constant, and when it changed.

  1.  The results section seems just to be a list of things which should be in a table. (That will also make it easier to see at a glance what increased and what decreased). It is tedious to read, and there doesn't seem to be much synthesis there. Placing all the results in the table would make it easier to see the increases and decreases at a glance. 

RE: We have added a table to facilitate a quick view of the tumor sites that are increasing and decreasing, compared with the Italian data.

  1. The discussion seems to be as disjointed as the rest of the manuscript. It is often unclear how the references support the argument. For example, the authors say "The increase among young individuals who frequently tan highlights the need for preventive measures [51]." The referenced paper does not seem to contain anything about tanning. They do mention lower odds of survival for 15-25 year olds, but from the analysis here, it is not clear whether the increase observed in this age group occurred in that age group or older. It is also not clear whether the authors ascribe the increase in tanning or something else.

RE: Thanks to the reviewer for the comment. We have rewritten the discussion trying to better justify what we observed and revising the bibliographical notes.

  1. Regarding the observed decline in thyroid cancer rates, the authors comment: "This decline may be linked to local efforts to reduce unnecessary diagnostic procedures, highlighting the effectiveness of regional public health measures in controlling cancer rates." But, if so, why isn't this observed in the rates for other cancers?

RE: We have better specified this part and used the same approach to better explain what happens for other tumor sites.

  1. The abstracts needs to be rewritten. The authors say "Our results indicate that young cancers affect 63% of women." And then the immediate next sentence is  "In particular, we observed in males a modest increase in the period 1996-2013 (APC 1.6*)"  .This reads like a complete non sequitur. The abstract goes on to list various trends, and then there is an abrupt conclusion that "These results underscore the importance of promoting healthy lifestyle behaviors among younger generations.", which, again, does not seem to follow from anything said before. 

RE: We have rewritten the abstract to better report data and comments.

  1. Also, in Line 1 of the Introduction, the authors say: "...cancer accounts for approximately 20 million new cases ..." Should this be "annually"? And again, 197x100,000 per year?

RE: Thanks for the comment. We have corrected the text and integrated with other references.

Reviewer 3 Report

Comments and Suggestions for Authors

I read with great interest the paper 'Population-Based Analysis of Cancer Incidence in Individuals Under 50 in a Northern Italy Province: Focus on Regional Disparities and Public Health Implications,' by Lucia Mangone et al. The manuscript reports on regional cancer trends among people under 50 in the province of Reggio Emilia, Italy. The authors conclude that overall, unlike international concerns about rising cancer rates among young people, their findings indicate only a modest increase in specific cancer types in their territory. These trends suggest to them that awareness campaigns on smoking cessation, healthy lifestyles, and adherence to oncological screening, if strongly pursued as in the Reggio Emilia territory, are effectively influencing cancer incidence, highlighting the essential role of localized public health strategies.

Although the paper is well-written and methodologically sound, I find it inadequate, if not methodologically incorrect, to correlate the reported data with hypothetical causes that, while possible, the study cannot prove. There is no proven relationship between the various ongoing prevention campaigns in that small geographical area and the reported epidemiological data. Moreover, given that in Italy, there is the AIRTUM association, which unites all existing cancer registries in the country, I believe it would be more appropriate to refer to that data collection rather than reporting data from a very limited area. Other minor notes concern the need for greater accuracy in the background literature data: before reference number 2, other epidemiological data describing the trend of cancer pathologies before the age of 50 are published (from which reference 2 is derived). The reported 'advancement of colorectal cancer screening to begin at age 45' is actually only applied in the United States, not in Europe.

Author Response

Comments and Suggestions for Authors

I read with great interest the paper 'Population-Based Analysis of Cancer Incidence in Individuals Under 50 in a Northern Italy Province: Focus on Regional Disparities and Public Health Implications,' by Lucia Mangone et al. The manuscript reports on regional cancer trends among people under 50 in the province of Reggio Emilia, Italy. The authors conclude that overall, unlike international concerns about rising cancer rates among young people, their findings indicate only a modest increase in specific cancer types in their territory. These trends suggest to them that awareness campaigns on smoking cessation, healthy lifestyles, and adherence to oncological screening, if strongly pursued as in the Reggio Emilia territory, are effectively influencing cancer incidence, highlighting the essential role of localized public health strategies.

Although the paper is well-written and methodologically sound, I find it inadequate, if not methodologically   1. incorrect, to correlate the reported data with hypothetical causes that, while possible, the study cannot prove. There is no proven relationship between the various ongoing prevention campaigns in that small geographical area and the reported epidemiological data.

RE: Thanks to the reviewer for the comment. We have rewritten the discussion better by reporting the data that we have observed and making only possible hypotheses that have been adopted directly by our Health Authority at a local and regional level.

2.Moreover, given that in Italy, there is the AIRTUM association, which unites all existing cancer registries in the country, I believe it would be more appropriate to refer to that data collection rather than reporting data from a very limited area. Other minor notes concern the need for greater accuracy in the background literature data: before reference number 2, other epidemiological data describing the trend of cancer pathologies before the age of 50 are published (from which reference 2 is derived).

RE: Thanks to the reviewer for the suggestion that we appreciate very much. Unfortunately in this case we are not able to involve all the Italian Cancer Registry to participate in the project. It is not excluded that in the future works will be published on a regional or national scale: we have made an effort to work on recent data and try to bring our experience to such a sensationally current fact as the increase in tumors in young people. We hope that the effort will be appreciated anyway and that we are not penalized because we are a small CR. We have added this aspect better within the limits of the study.

Thank you, we have added other citations in the introduction.

3.The reported 'advancement of colorectal cancer screening to begin at age 45' is actually only applied in the United States, not in Europe.

RE: Thanks for the comment. We have fixed this in the text.

Round 2

Reviewer 1 Report

Comments and Suggestions for Authors

The suggestions have been added to the manuscript. Overall, good manuscript.

Author Response

Comments and Suggestions for Authors

The suggestions have been added to the manuscript. Overall, good manuscript.

RE: We thank the reviewer for the positive evaluation of the manuscript.

Reviewer 2 Report

Comments and Suggestions for Authors

I feel that some aspects of the paper have improved, others have actually deteriorated. The rewritten discussion seem to have a lot of badly phrased confusing sentences.  My comments are below:

In paragraph one of the introduction, the authors probably meab 196.9/100,000 per year rather than 196.9 x 100,1000 ?

The paper needs to be edited for clarity. There are a lot of words which makes for longer sentences but obfuscate the meaning.

In Data analysis section, 1st sentence, SIRs were not "developed" they were "calculated". In the 2nd sentence you say:

"The demographic estimates for the derivation of the rates are based on the registered population..."

It is unclear what it actually means. Did you just use the population from the population registry for the specific years? Or did you interpolate/extrapolate the population somehow? That needs to be explained.

You do not need to mention the explicit logarithmic transformation, since it is unclear what input file tab you are referring to, and what are y and b and x. It is sufficient to say that you used a segmented log-linear regression as is customary etc (as per your response letter). This justification and the references to other papers/reports that use the same methods need to be included in the manuscript, not just in the response letter.

I also find it confusing that the authors say in the manuscript that the analysis has been dones using Stata, but in the response letter seem to refer to some other software. If it is a package for Stata, it should be reference in addition to referencing Stata.

My comments about graphs were not due to my lack of understanding of how log-linear models work. Rather they were due to the fact that the model is not shown in the graph. Please add the estimated trends and 95% confidence envelopes to the graphs to demonstrate. And please increase the font size for the axes labels and other annotations to improve readability. 

In Table 4, why are some cells empty and other 'na'? I thought initially empty cells were for non-existing cancers (such as uterus in males), but I am pretty sure females have pancreas.

Line 301,

The phrase "Great attention in our country also to aspects linked to environmental pollution" is grammaically incorrect and lacks a verb.

The sentence in lines 307-309 starts with a point, and is also rambling and grammatically incorrect. What does "on the stage" mean?

In line 313, the words "in our experience" are superfluous.

In line 319, please rephrase "implement the physical activity".

Comments on the Quality of English Language

Discussion in particular needs to be carefully edited. Please see my comments above.

Author Response

Thanks for your comments, we hope we have answered all your questions exhaustively.
Best regards,
Isabella Bisceglia
